# The Influence of Different Factors on the Metabolism of Capsaicinoids in Pepper (*Capsicum annuum* L.)

**DOI:** 10.3390/plants13202887

**Published:** 2024-10-15

**Authors:** Yuanling Yang, Chengan Gao, Qingjing Ye, Chenxu Liu, Hongjian Wan, Meiying Ruan, Guozhi Zhou, Rongqing Wang, Zhimiao Li, Ming Diao, Yuan Cheng

**Affiliations:** 1State Key Laboratory for Managing Biotic and Chemical Threats to the Quality and Safety of Agro-Products, Vegetable Research Institute, Zhejiang Academy of Agricultural Sciences, Hangzhou 310021, China; yuanlingyang@outlook.com (Y.Y.); gaochengan@126.com (C.G.); yeqj@zaas.ac.cn (Q.Y.); liuchenxu@zaas.ac.cn (C.L.); hjwan@zaas.ac.cn (H.W.); ruanmy@zaas.ac.cn (M.R.); zhougz@zaas.ac.cn (G.Z.); wangrq@zaas.ac.cn (R.W.); zhimiaoli@zaas.ac.cn (Z.L.); 2College of Agriculture, Shihezi University, Shihezi 832003, China; 3College of Horticultural Sciences, Zhejiang A&F University, Hangzhou 311300, China

**Keywords:** pepper, capsaicinoids, metabolism pattern, different factors

## Abstract

Pepper is a globally cultivated vegetable known for its distinct pungent flavor, which is derived from the presence of capsaicinoids, a class of unique secondary metabolites that accumulate specifically in pepper fruits. Since the accumulation of capsaicinoids is influenced by various factors, it is imperative to comprehend the metabolic regulatory mechanisms governing capsaicinoids production. This review offers a thorough examination of the factors that govern the metabolism of capsaicinoids in pepper fruit, with a specific focus on three primary facets: (1) the impact of genotype and developmental stage on capsaicinoids metabolism, (2) the influence of environmental factors on capsaicinoids metabolism, and (3) exogenous substances like methyl jasmonate, chlorophenoxyacetic acid, gibberellic acid, and salicylic acid regulate capsaicinoid metabolism. The findings of this study are expected to enhance comprehension of capsaicinoids metabolism and aid in the improvement of breeding and cultivation practices for high-quality pepper in the future.

## 1. Introduction

Pepper (*Capsicum annuum* L.), a prominent member of the *Solanaceae* vegetable family, is extensively grown globally for its distinctive flavor and nutritional richness, encompassing capsaicinoids, vitamins, and carotenoids [1,2,3,4]. The International Board for Plant Genetic Resources (IBPGR) has identified five distinct pepper cultivars, namely *Capsicum annuum*, *Capsicum frutescens*, *Capsicum chinense*, *Capsicum baccatum*, and *Capsicum pubscens*. Capsaicinoids, identified exclusively in *Capsicum*, are metabolites responsible for the distinct pungency flavor and have been associated with various health benefits, such as antioxidant [5], anti-cancer [6,7], and anti-inflammatory properties [8]. Additionally, capsaicinoids have diverse applications in fields such as feeding [9], military [10], etc. Currently, a total of 23 types of capsaicinoids have been identified [11], with capsaicin and dihydrocapsaicin being the most prominent, comprising approximately 69% and 22% of the total capsaicinoid content [11,12,13], respectively. The variations in capsaicinoids are primarily linked to the structural characteristics of the side-chain aliphatic hydrocarbons, including their length, presence of double bonds, and branching points [14]. From an evolutionary perspective, the biosynthesis and accumulation of capsaicinoids have been driven by their role in deterring mammalian feeding behaviors [15,16,17] and inhibiting *Fusarium* infections [18].

Interest in capsaicinoid research has been significant across various academic disciplines since the initial isolation of impure capsaicinoid crystals from peppers by German chemist Christian Friedrich Bucholz (1770–1818) through solvent extraction in 1816 [19]. In recent years, David Julius and Ardem Patapoutian were awarded the Nobel Prize in medicine for their groundbreaking research on the capsaicinoid receptor *TRPV1*. Since 2014, research teams worldwide have extensively studied the *Capsicum* genome [20,21,22], leading to a significant increase in investigations into the regulatory pathways of capsaicinoid metabolism and their response to diverse environmental conditions [23,24,25,26].

To date, research has demonstrated that capsaicinoids are synthesized by glandular cells within the placenta and pericarp tissue of pepper fruit through the phenylpropanoid pathway and branched-chain fatty acid pathway [14]. Over 50 genes have been identified as playing a role in the biosynthesis of capsaicinoids [14,27,28]. The regulation of capsaicinoid biosynthesis involves transcription factors from various families, such as *MYB31* from the *MYB* family, *WRKY09* from the *WRKY* family, and Erf and Jerf from the *AP2/ERF* family, among others [23,29,30,31,32]. The level of capsaicinoids in pepper is subject to dynamic equilibrium as a result of competition for their common substrate with other pathways, such as flavonoids, tannins, and lignin. Additionally, capsaicinoids are known to be unstable and susceptible to degradation by peroxide in the presence of H_2_O_2_, although the specific mechanism of this degradation remains unclear [33]. Research has shown that capsaicinoid accumulation is influenced not only by the pepper genotype and fruit maturation stage [27,34], but also by various environmental factors, including light [30], temperature [35], soil nutrients [36], water [37], hormones [38], and pathogens [20]. These factors collectively govern the biosynthesis of capsaicinoids [39], a quantitative trait that is influenced by various environmental factors [20,40].

This paper provides a comprehensive overview of existing research on the effect of biotic/abiotic stresses and genotypes on capsaicinoid metabolism, as well as the transcriptional regulatory mechanisms involved in capsaicinoid metabolism. The aim is to offer theoretical insights into the analysis of the regulatory mechanisms governing capsaicinoid production and suggest potential advancements in the nutritional and flavor quality of chili pepper cultivation and storage techniques.

## 2. Capsaicinoid Metabolism Differed in Different Genotypes

Capsaicinoid content in pepper is predominantly influenced by their genotypes [41], with pepper being categorized into five domesticated species (*C. annuum*, *C. baccatum*, *C. chinense*, *C. pubescens*, and *C. frutescens*) that exhibit significant variation in capsaicinoid levels [13,42]. In order to investigate the correlation between capsaicinoid content and different cultivars, all peppers are cultivated in a uniform environment to mitigate the effect of external factors. A comprehensive analysis of 136 cultivars revealed that *C. chinense* exhibited the highest capsaicinoid content, while *C. annuum* displayed the lowest levels [43]. Furthermore, the expression of capsaicinoid biosynthesis genes (*CBGs*) exhibited variability across different pepper varieties, with expression levels showing a positive correlation with capsaicinoid content [43]. The comprehensive overview of genetic resources in *capsicum* pepper provides valuable insights into the process of pepper breeding selection and identification. Korean scholars utilized genome-wide association study (GWAS) analysis to investigate the genetic underpinnings of carotenoid and capsaicinoid content within a collection of 160 *C*. *chinense* germplasms. The findings of this study indicate that single nucleotide polymorphisms (SNPs) have the potential to serve as a tool for selecting pepper germplasm with elevated levels of carotenoids and capsaicinoids [41].

The earliest studies found that capsaicin biosynthesis involves two major pathways, the phenylpropanoid and branched chain fatty acid pathways [14], which start with phenylalanine and valine, respectively, and gradually convert these amino acids into the key prerequisites for capsaicin synthesis (vanillylamine and 8-methyl-6-nonenoyl-CoA) catalysed by a series of enzymes, which are ultimately condensed to form capsaicinoids under the action of CS (capsaicinoid synthase). The pathway of capsaicin biosynthesis is shown in Figure 1. To investigate the evolutionary history of the capsaicinoid biosynthesis pathway, a recent study reconstructed the phylogenetic tree of capsaicinoids. The capsaicinoid synthesis pathway was found to occur between approximately 13.4 million years (*C. rhomboideum* and *C. annuum*/*C. baccatum*) and 5 million years (*C. baccatum* and *C. annuum*). The study analyzed the evolution of *CBGs* in peppers and found that all peppers, whether they produce capsaicinoids or not, contain different amounts of *CBG* genes [44]. Researchers discovered that hybrid offspring exhibited markedly elevated levels of capsaicinoid when compared to their parental genotypes [45], providing evidence for the heterosis phenomenon in secondary biosynthesis, specifically in capsaicinoid biosynthesis. An average rise of 65–72% in the expression of certain metabolites and genes associated with metabolite biosynthetic pathways was noted in F_1_ hybrid descendants as opposed to their progenitors [46]. These findings indicate that the secondary metabolism of F_1_ offspring, particularly in capsaicinoid biosynthesis, may display a beneficial heterosis effect [47]. In conclusion, the capsaicinoid content in peppers is predominantly influenced by their genotypes, exhibiting notable variation across various cultivars and species. Although harvest timing and environmental conditions play important roles in determining the levels of these compounds, meticulous experimental planning can mitigate these variables and facilitate the elucidation of the genetic determinants and pathways involved in capsaicinoid biosynthesis in pepper. Furthermore, the potential of hybrid progeny presents an optimistic prospect for enhancing the capsaicinoid content in peppers.

The genetic background predominantly influences the synthesis of capsaicinoids [48]. Classical genetic research has revealed that capsaicinoid synthase activity is governed by the dominant *pun1* gene, which facilitates the ultimate stage of capsaicinoid production, thereby influencing the presence or absence of capsaicinoids [49]. Additionally, investigations have indicated that a majority of sweet pepper varieties (*Capsicum annum* L.) exhibit a comparable deletion of the *pun1* locus [50]. Four mutations in pun1 have been identified to date: *pun1^1^*, *pun1^2^*, *pun1^3^*, and *pun1^4^* [50,51,52]. These findings provide valuable insights into the genetic regulation of capsaicinoid biosynthesis in peppers and suggest potential avenues for future research to explore the relationship between genotype and capsaicinoid content in various pepper cultivars.

Capsiate, an alkaloid similar to capsaicin known as 4-hydroxy-3-methoxybenzyl (E)-8-methyl-6-nonenoate, was discovered in Sweet CH-19 peppers, which are non-pungent [53]. Capsaicinoid is an amide of vanillin and various branched-chain fatty acids, while capsiate is the ester of vanillin and a fatty acid [54,55]. The process of capsaicinoid biosynthesis actively relies on the presence of pAMT, which is a key gene that participates in the phenylalanine metabolic pathway. The absence of pAMT hinders the production of vanillylamine, ultimately resulting in the loss of pungency [56]. Nine mutants of pAMT have been identified, including single nucleotide substitutions, single and multiple nucleotide insertions, and transposon insertions that result in the inability of vanillin to form vanillamine [51]. Interestingly, this loss of pungency leads to the formation of capsiate. In the absence of capsaicinoid aminotransferase, vanillin accumulates and is subsequently converted to vanillylamine by cinnamyl alcohol dehydrogenase CAD, ultimately catalyzed by capsaicinoid synthase to synthesize capsiate [57,58].

The regulation of capsaicinoid biosynthesis genes is influenced by various transcription factors, with *MYB31* identified as a key transcription factor that interacts with the promoter region of *CBGs* to modulate capsaicinoid biosynthesis. Deletion of MYB31 leads to reduced CBG expression, which reduces capsaicinoids, such as capsaicin and dihydrocapsaicin [30]. A previous study found that MYB31 was able to regulate the expression of the capsaicin biosynthesis pathway structural genes *BCAT*, *C4H*, *4CL*, *COMT*, *pAMT*, *FatA*, *KAS*, *ACL*, and *CS* [30]. It was also able to regulate the presence of a w-box in the *MYB31* promoter region, which is specifically recognised and activated by the *WRKY9* transcription factor in the placenta [32]. This activation leads to high levels of transcription of *MYB31* and robust activation of *CBGs*, ultimately leading to the synthesis of large amounts of capsaicin. Thus, MYB31 may be a major regulator of capsaicin biosynthesis. In the *bHLH* family, the transcription factors *bHLH7, 9, 26, 63,* and *86* are involved in capsaicin biosynthesis by interacting with *MYB31* [59]. Recent studies have found that *MYB24, 4* negatively regulates capsaicin biosynthesis [60,61]. The downregulation of the *AT3* gene in peppers leads to decreased expression of genes *pAMT*, *BCAT*, *KAS*, and *ACL,* which are involved in capsaicinoid biosynthesis [30]. Additionally, the transcription factor *CaMYB48* has been identified as a key player in capsaicinoid biosynthesis. Sun et al. utilized weight gene co-expression network analysis (WGCNA) and fruit placenta transcriptome data to demonstrate the involvement of *CaMYB48* in capsaicinoid biosynthesis [62]. Furthermore, *CaERF2*, which is induced by ethylene, is implicated in the regulation of *CBGs* and may serve as a signal for ripening [63].

*CaMYB37*, a transcription factor, has recently been identified as a significant contributor to capsaicin biosynthesis by regulating *CBG* expression through binding to the *AT3* promoter region, similar to *CaMYB31* [64]. *MYB37* was able to regulate the expression of the structural genes of the capsaicin biosynthesis pathway, *pAMT*, *KAS*, *ACL*, and *CS*, suggesting that it is also one of the key regulators of capsaicin biosynthesis. *C. chinense*, such as the “Carolina Reaper” and “Butch T” varieties (www.guinnessworldrecords.com), are known for their extremely high capsaicinoid content. Previous assumptions suggested that capsaicinoid synthesis was limited to the placenta [48]. Nevertheless, recent research has revealed that substantial quantities of *CBGs* are present in the pericarp of “MY” chili peppers, challenging prior assumptions [65]. Additionally, Sun’s recent study has provided evidence of *MYB31*’s involvement in the regulation of capsaicinoid synthesis in the pericarp [66].

Related genes regulating capsaicinoid synthesis are being explored [28,44,52]. Some teams also tried to search for genes related to capsaicinoid synthesis through mutants, and this study used ethyl methanesulfonate (EMS) mutants to locate the *Pun4* site. Genetic, genomic, and transcriptome analysis indicated that *Pun4* might be involved in capsaicinoid synthesis [67].

**Figure 1 plants-13-02887-f001:**
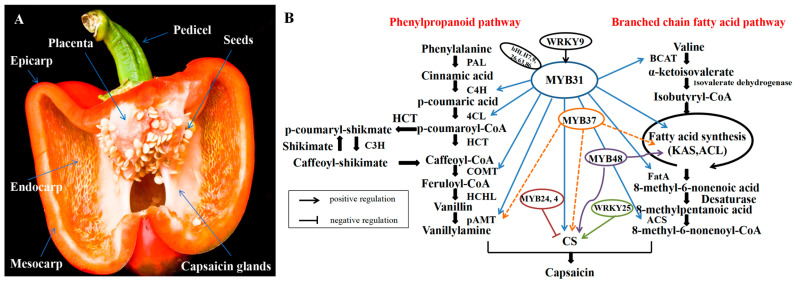
Capsaicinoid biosynthetic pathway. (**A**): Cross-section of chili fruit; (**B**): schematic diagram of capsaicin biosynthesis pathway and regulatory genes. Abbreviations: PAL, phenylalanine ammonia lyase; C4H, coumarate 4-hydroxylase; 4CL, 4-coumaroyl-CoA ligase; HCT, hydroxycinnamoyl transferase; C3H, coumarate 3-hydroxylase; COMT, caffeoyl-CoA 3-O-methyltransferase; HCHL, hydroxycinnamoyl-coenzyme A hydratase lyase; pAMT, Putative aminotransferase; BCAT, branched-chain amino acid aminotransferase; KAS, ketoacyl-ACP synthetase; ACL, acyl carrier protein; FatA, acyl-ACP-thiesterase; ACS, acyl-CoA synthetase; CS, capsaicinoid synthase; WRKY (WRKY9, 25), bHLH (bHLH7, 9, 26, 63 and 86), MYB (MYB4, 24, 31, 37 and 48) are the genes that have been shown in current studies to be involved in the capsaicin regulatory pathway, adapted from Naves et al. [20] and QIN et al. [68]. Note: Refer to Aza-Gonzalez et al. [12] for the chemical structure of capsaicin synthesis pathway-related substances. Refer to Zhang et al. [69]and Naves et al. [20] for the pictures of germplasm resources of different varieties.

## 3. The Metabolism of Capsaicinoids in Different Developmental Stages of Pepper Fruits

The fluctuation of secondary metabolite levels in fruits during their maturation is attributed to the complex interplay of various metabolic pathways [70]. For instance, capsaicinoid levels were observed to be absent at 20 days post-anthesis (dpa), reaching a peak at approximately 40 dpa and subsequently declining. Although capsaicin content generally rises as fruits ripen, it diminishes post-full ripeness, as evidenced by the research of Salvador [71]. Kim et al. [27] conducted an analysis of capsaicinoid synthesis-related genes in “CM334” chili variety, revealing that the expression of capsaicin synthase genes commenced early in fruit development, decreased as the fruit matured, and ceased by day 10 of the color change stage. Additionally, Mercedes Vázquez Espinosa observed fluctuations in capsaicinoid content among certain varieties, with an initial increase followed by a sharp decline of 90% by day 40 after anthesis and subsequent recovery on some level [72].

Capsaicinoids may undergo degradation during the development of fruit. Research indicates that peroxidase plays a role in the biosynthesis of capsaicinoids and the breakdown of capsaicinoids, potentially through the oxidation of phenolic precursors of capsaicinoids, as noted by Contreras M et al. [73] and DJ et al. [74]. Bernal et al. [75] showed that the products of capsaicin oxidation by peroxidases are 5–5′-dcapsaicin, 4′-O-5-dicapsaicin ether, and some highly polymerized dehydrogenation products. Barrera et al. [76] did not find a negative correlation between peroxidase and capsaicin, possibly due to the fruit not yet undergoing the aging process. Palma et al. [77] discovered that antioxidants such as ascorbic acid and glutathione may play a role in preserving capsaicinoid biosynthesis and enhancing their concentration. Prior studies have shown that ascorbic acid can efficiently diminish the free radicals produced by capsaicinoid substances through peroxidase catalysis. Additionally, the capsaicinoid content of chili fruits within the same variety displayed fluctuations at various harvest times, with the capsaicinoid content of “Havana pepper” following a normal distribution pattern dependent on harvesting time, while other chili peppers exhibited a skewed distribution. This observation suggests a potential association with both genotype and harvest timing [42]. However, the comprehensive mechanism governing the regulation of capsaicinoid metabolism remains ambiguous and necessitates additional exploration.

## 4. Regulation of Different Environmental Factors on the Metabolism of Capsaicinoids

Environmental factors also regulate capsaicinoid metabolism (Figure 2). Light plays a crucial role in regulating capsaicinoid metabolism in chili peppers by impacting gene structure and enzyme activities [30,31]. Temperature is another significant factor that influences capsaicinoid metabolism. In the context of chili pepper cultivation, elevated temperatures have been shown to enhance capsaicinoid levels in both bell peppers and chili peppers [35]. Furthermore, environmental variables, including drought [37], salinity [78], and nitrogen [36] availability, play crucial roles in the synthesis of capsaicinoids. Additionally, biological stress [21] and mechanical injury [20] are implicated in the regulation of capsaicinoid metabolism, with hormones serving as key regulators in this process. In chili peppers, methyl jasmonate (MeJA) [4], gibberellin (GA3) [38], and salicylic acid (SA) [79] have been shown to increase the expression of genes involved in capsaicinoid biosynthesis, including *CaMYB108* and *CBGs*.

### 4.1. Temperature

Temperature has been shown to influence the expression of *CBGs*, thereby impacting the accumulation levels of capsaicinoids in various pepper varieties. The content of capsaicin and dihydrocapsaicin in peppers such as “Serrano”, “Puya”, “Anchoro”, “Guajillo”, and lantern pepper has been observed to increase with rising temperatures, while other varieties exhibit a contrasting trend [35]. Numerous studies have demonstrated that high-temperature treatment leads to an increase in capsaicinoid content [35,80]. In the study conducted on pepper plants, exposure to high temperatures followed by a decrease in temperature led to a pattern of initially decreasing and then increasing capsaicinoid levels [81]. A comparable investigation on low-pungent pepper varieties yielded analogous findings [82].

The profile of fruit metabolites undergoes alterations throughout the stages of fruit development, ripening, and postharvest handling, encompassing capsaicinoids. Additionally, the composition of fruit metabolites postharvest undergoes further modifications during storage. The capsaicinoid content of postharvest fruits exhibited variability in relation to storage duration and temperature. After 18 days of storage, the capsaicinoid content was found to increase under the room-temperature (25 °C) storage condition compared to the low-temperature (0 °C) storage condition [83]. Previous short-term storage experiments have also demonstrated an increase in capsaicinoid levels under high temperatures compared to the control [80]. Furthermore, the changes in capsaicinoid content during short-term storage closely mirror those observed during pre-harvest fruit maturation, potentially due to the preservation of CS activity. Different fruit-drying treatments may lead to fluctuations in capsaicinoid content. The process of drying fresh pepper at temperatures ranging from 45 °C to 65 °C reduces capsaicinoid and dihydrocapsaicin levels by over 30%, and the treatment at 45 °C preserves more capsaicinoids. [84]. Recent research conducted by Grimaldi et al. supports these findings [22]. The decrease in capsaicinoid content during the drying process may be attributed to the catalytic alteration of its chemical structure at elevated temperatures (~120 °C) [85], or, alternatively, to the oxidation reaction catalyzed by peroxidase at temperatures below the high range (~60 °C) [86]. In contrast, a recent study demonstrated a 6-fold increase in capsaicinoids following hot air-drying treatment (120–160 °C), accompanied by a decrease in total flavone content and an increase in total phenol content (38–51%) [87].

Kim et al. [88] conducted experiments involving the fusion of *CS* promoter (*pun1*) with the β-glucuronidase (*GUS*) gene, which was then introduced into the *Arabidopsis* genome. Their findings indicated that seedlings subjected to high temperatures (37 °C) exhibited a 98.61% increase in *GUS* activity compared to those treated under room temperature (23 °C). Additionally, high-temperature treatment (40 °C) was shown to enhance the expression of the *pPUN1:GUS* gene in tobacco [89]. The expression of *CaMYB31*, *KAS*, and *pAMT* in “Serrano ‘Tampiqueño 74’” peppers was significantly increased when exposed to 4 °C and decreased at 37 °C compared to 25 °C treatment. Fan et al. [90] showed that DnaJ (Hsp40) proteins play crucial roles in plant growth and development and were responses to abiotic stress. They analyzed eight *CaDnaJ* genes and found that the expression of *CaDnaJ74* was significantly upregulated under high-temperature stress, and the co-expression of *CaDnaJs* with *CBGs* was observed, indicating a possible role of *CaDnaJ74* in capsaicinoid synthesis regulation under high temperatures.

### 4.2. Water

Water stress has been shown to affect plant growth and development, resulting in alterations in capsaicinoid content. Phimchan’s [37] research indicated that drought treatment led to an increase in capsaicin and dihydrocapsaicin content and a decrease in yield for low-spicy pepper varieties, while high-spicy varieties displayed no significant change, possibly due to their adaptation to hot climates. It is suggested that high-spicy and small fruit pepper varieties may exhibit greater tolerance to drought conditions, with less impact on their yield and capsaicinoid content. Nevertheless, in cases of severe drought, capsaicinoid content in high-spicy pepper varieties may still rise [91]. The equilibrium between capsaicin and other phenylpropanoid pathways is disrupted under drought stress, resulting in fluctuations in capsaicin content [92]. While optimal drought stress conditions may enhance capsaicinoid accumulation, the response varies among different varieties and genotypes [93]. Various studies have demonstrated that moderate drought conditions can elevate capsaicinoid levels, whereas severe drought may decrease them [24,94]. Peng’s [94] research revealed that capsaicinoid content initially decreased compared to the control group after 40 days of mild drought treatment but subsequently increased after 60 days. In their study, Haak et al. [95] noted a negative relationship between the quantity of pepper seeds per fruit and capsaicinoid levels under drought treatment. Additionally, it has been demonstrated that capsaicinoid levels are notably diminished in chili peppers exposed to drought stress during the nutritional phase in comparison to the flowering and fruiting phases [96]. Consequently, ensuring adequate water supply during the nutritional phase of chili cultivation is essential for enhancing both yield and fruit quality. On the other hand, in contrast to commercial irrigation (CI), deficit irrigation may lead to a decrease in pepper yield while increasing the concentration of soluble solids and capsaicinoids in fruit, potentially enhancing drought resistance through their role as osmotic protectants [97].

The phenylpropane pathway and the synthesis of secondary metabolites are notably affected by water stress. Research has shown that drought conditions can elevate the activity of capsaicinoid biosynthetic enzymes such as *C4H*, *CS*, and *PAL* while also reducing capsaicinoid degradation by suppressing peroxidase activity [37,73]. In contrast to the effects of overwatering, drought stress resulted in a notable upregulation of *CaMYB31* and *WRKY9* expression levels in the “Shishito” and “Sapporo OonagaNanban” cultivars, leading to the activation of various target genes such as *KAS*, *pAMT*, *Pun 1*, *FAT*, and *BCAT* [26]. Furthermore, aside from capsaicinoid biosynthetic enzymes, DnaJ proteins have been implicated in their response to drought stress. Specifically, five *CaDnaJ* genes (*CaDnaJ10, 25, 46, 47,* and *56*) were significantly upregulated and exhibited high co-expression with *CBGs* under drought conditions [98]. The accumulation of capsaicinoids can be influenced by overwatering, with variations in genotypes, as well as ecological and physiological factors [20]. The impact of overwatering on capsaicinoid content may differ among pepper varieties [20]. In some cases, excessive watering in certain pepper varieties may lead to no alteration or a reduction in capsaicinoid levels [26]. Moreover, capsaicinoids were identified as a natural defense mechanism against pathogenic bacteria, especially in environments with high humidity conducive to fungal growth [95].

### 4.3. Light

The significance of light in plant physiology is paramount, as it serves as the primary energy source for photosynthesis and as a regulatory signal for plant growth. Furthermore, light has also proven to be crucial in the regulation of capsaicinoid metabolism [70]. The effect of varying light intensities on capsaicinoid content has been found to be inconsistent in different literature. Ombódi et al. [99], Gao et al. [100], and Lv et al. [101] conducted studies that revealed an increase in capsaicinoid levels under shade treatment. Specifically, the highest levels of capsaicinoid were observed in high-spicy variants under 50% and 70% shading, while the lowest levels were found in plants exposed to full light [102]. Conversely, another study indicated that excessive shading led to a decrease in capsaicin content, as well as fruit yield, likely due to reduced light intensity and subsequent lower photosynthetic efficiency [103]. Therefore, it can be inferred that achieving an optimal light intensity is essential for increasing the capsaicinoid content in chilies. Additionally, research has shown that exposing pepper to light can enhance capsaicinoid accumulation in the pericarp part, whereas storing them in dark conditions can increase capsaicinoid levels in the placenta [104].

Various types of light, including UV-C and blue light, both play roles in regulating capsaicinoid biosynthesis by inducing the synthesis of flavonoids and providing protection against oxidative stress as abiotic stresses [105,106]. Multiple studies have demonstrated that the use of LED blue light treatment can stimulate the ripening process [107]. These observations are supported by the research conducted by Gangadhar et al. [108], which highlights the potential application effects of blue LED lights in various pepper cultivars. For example, the experiments of blue light and UV-B radiation on “habanero” pepper (*Capsicum chinense*) fruit over different durations suggested that blue light treatment can induce the accumulation of chlorophyll b, total flavonoids, capsaicinoids, and other antioxidant properties. The capsaicinoid content in most pepper fruits tends to rise when exposed to blue light or UV-C light during storage, leading to a noticeable enhancement in the antioxidant capacity of peppers under blue light and UV-C light treatment [102]. Levels of capsaicinoid in postharvest peppers can be elevated through exposure to red and blue light, while essential amino acids and aromatic amino acid levels can be heightened by exposure to white and red light [109]. Furthermore, dark storage seems to be a more favorable condition for maintaining capsaicinoid levels of pepper compared to those treated by LED blue light [110]. Exposure to blue light results in a shift in the precursor to capsaicinoids, with a decrease in phenylalanine and an increase in cinnamic acid, potentially accelerating the biosynthesis of capsaicinoids [111]. The expression of genes involved in capsaicinoid biosynthesis, including *CS*, *KAS*, and *pAMT*, is positively regulated by light due to the presence of photosensitive elements in the promoter region, leading to enhanced capsaicinoid production [30]. Light affects the transcription factor *CaMYB1,* which regulates capsaicinoid synthesis genes. The expression level of *CaMYB1* is upregulated under light compared to dark conditions. Similarly, *KAS* and *pAMT* exhibit a similar expression trend [30]. Additionally, light-induced peroxidase has also been proven to indirectly decrease capsaicinoids [74].

### 4.4. Soil Nutrition

Different soils affect capsaicinoid synthesis. A study conducted by Mexican researchers examined the impact of black and red soils on the genetic regulation of capsaicinoid synthesis. The findings indicated that peppers cultivated in red soil exhibited increased fruit size, whereas peppers grown in black soil enhanced capsaicinoid biosynthesis [112]. Significantly higher capsaicinoid content and capsaicinoid synthase activity were observed in peppers cultivated in alluvial soil compared to those grown in lateritic soil. This difference was reflected in the upregulation of the gene *Csy1*, which encodes capsaicinoid synthase, in peppers from alluvial soil. The upregulation of the *pun1* gene in red soil was found to enhance pungency, while the overexpression of the recessive gene *pun12* was observed to suppress capsaicinoid content. Optimal soil nutrition is crucial for the development of highly pungent fruit [113].

Competitive relationships are recognized to exist among various synthesis pathways, with fertilization being a factor that disrupts nutrient balance and leads to alterations in capsaicinoid content. Naves et al. [20] proposed that nitrogen plays a key role in the biosynthesis of capsaicinoids. The biosynthesis of capsaicinoid involves phenylalanine, valine, and leucine, along with an unidentified amino donor necessary for vanillin synthesis. The levels of capsaicinoids are significantly influenced by the interactions of pepper genotype and soil nutrients [114]. Multiple studies demonstrated a positive correlation between capsaicinoid contents and soil nitrogen levels [115,116]. Furthermore, experiments conducted using hydroponics have shown that elevated nitrogen levels can enhance capsaicinoid production. Additionally, the application of nitrogen fertilizer has been found to promote pepper plant growth, increase fruit yield, and maintain capsaicinoid levels [36]. The capsaicinoid metabolism regulated by nitrogen proved to be nitrogen-type dependent. For example, Monforte Gonzalez et al. [36] conducted a study comparing various NH_4_^+^: NO_3_^−^ ratios and determined that the 25:75 ratio led to a significant increase in capsaicinoid levels, as well as the activation of GOGAT (glutamine oxoglutarate aminotransferase) and GS (glutamine synthetase) enzymes, which are associated with the enhanced synthesis of capsaicinoid in pepper fruits [117]. Except for nitrogen, capsaicinoid metabolism was also demonstrated to be affected by other soil elements, as Da Silva et al. [118] observed that capsaicinoid content in pepper fruits increased in response to potassium deficiency, while it decreased under sulfur deficiency. Increasing potassium application has been found to significantly decrease capsaicinoids [116]. Additionally, the use of nigari, a byproduct of salt industries, has been shown to enhance capsaicinoid levels. Research indicates that a concentration of 4 mL·L^−1^ of nigari could result in 2.5-foldercapsaicinoid accumulation compared to the control group, possibly due to the mild salt stress, which leads to the upregulation of *CS* gene expression [119]. Other salt stress experiments also resulted in the increase of capsaicin and dihydrocapsaicin contents in the fruits of capsaicinoids [120]. Simultaneously, various soil compositions also influence the metabolism of capsaicinoids.

### 4.5. Pathogens and Wounding

The synthesis of capsaicinoids may serve as an adaptive mechanism in response to selective pressure from pathogenic bacteria [121]. Initial studies suggested that capsaicinoids in chili peppers can impede the growth of mycelium from chili blight pathogens (*Fusarium oxysporum*), thus enhancing plant resistance [104]. Recent studies have shown that capsaicinoids can inhibit ATP production by binding to NADH dehydrogenase, leading to the inhibition of oxidative phosphorylation and hindering the growth of pathogenic bacteria [21]. Additionally, capsaicinoids are found to provide protection against seed invasion in chili plants [122].

The concentrations of capsaicinoids in *capsicum* fruit can be induced by wound, which could upregulate the expression of capsaicinoid synthase-related genes and subsequently enhance the activities of CS and PAL enzymes [20,88,123]. Jasmonic acid (JA), a phytohormone, has been shown to be involved in capsaicinoid synthesis [62]. JA synthesis increases when chili is subjected to mechanical damage, thereby increasing capsaicinoid content [124]. Treatment with JA has been shown to significantly reduce the expression of *CaMYB31*, *KAS*, and *pAMT* in wounded *Capsicum* fruits [30]. However, there is a paucity of research on the influence of induced injury on capsaicinoid biosynthesis, necessitating further exploration.

### 4.6. Regulation of Different Exogenous Substances

Exogenous compounds may exert diverse influences on capsaicinoid metabolism. The utilization of methyl jasmonate on peppers resulted in a notable elevation in capsaicin and dihydrocapsaicin levels [125], a phenomenon ascribed to the upregulation of *CaMYB108* and *CBGs* [62]. Leaf application of chlorophenoxyacetic acid (CPA), Gibberellic acid (GA3), and naphthalene acetic acid (NHA) significantly improves fruit quality and capsaicinoid levels in diverse pepper varieties relative to untreated samples [38]. The administration of plant growth regulators has the potential to alter *CBG* expression, thereby influencing their function as signaling molecules. Following amino acid treatment with biological stimulants, the activities of PAL, CS, and POX in pepper leaves demonstrate a rapid initial increase within 1 h, gradually returning to pretreatment levels after 72 h. This phenomenon reflects the dynamic nature of capsaicinoid metabolism, influenced by various factors [126]. Additionally, the foliar application of H_2_O_2_ significantly enhanced the levels of total phenols, total flavonoids, and capsaicinoids while upregulating the expression of *PAL*, *pAMT*, and *KAS* [127]. Furthermore, the exogenous application of these substances during postharvest storage helped to maintain elevated capsaicinoid content. Pepper fruits treated with spermidine and putrescine exhibit a notable retention of capsaicinoids [128]. Additionally, the increased activity of antioxidant enzymes such as catalase (CAT), superoxide dismutase (SOD), and peroxidase on leaves sprayed with salicylic acid (SA) and coated with coriander oil after harvesting led to the enhancement of the free radical scavenging capacity of chili, which resulted in effective retention of capsaicinoid content [79]. The presence of reactive oxygen species in stored fruits can decrease capsaicinoids, but external treatments aid in mitigating this process by bolstering antioxidant levels [128]. Capsaicinoid suppresses capsaicinoid promoter activity and inhibits the expression of *KAS*, *PAL*, and *pAMT* in the presence of exogenous capsaicinoids, indicating its significant role in the feedback inhibition of capsaicinoid biosynthesis, as documented by Kim [88]. Additionally, research conducted by Derek W. Barchenger [129] demonstrated that exogenous capsaicinoids impede pepper seed germination.

In summary, it is clear that both appropriate biotic and abiotic stresses promote the synthesis of capsaicin, as well as other capsaicinoids. In fact, these stresses can also change the fruit size and yield of chili to some extent. For example, nitrogen fertiliser not only increases capsaicin content but also promotes larger fruit size, thus increasing chili yield [116]. Secondly, chili is a shade-loving crop, and proper shading can prevent chili from being burnt by the sun, thus promoting plant growth [102].

## 5. Problems and Prospects

The current understanding of the biosynthesis pathway of capsaicinoids reveals that the absence of *pAMT* and *pun1* functions impede the synthesis of capsaicinoids. However, further investigation is needed to confirm the impact of other genes associated with capsaicinoid synthesis on the spicy flavor. Additionally, the levels of capsaicinoids in pepper fruit exhibit notable variations across various developmental stages. From an epigenetic standpoint, it is imperative to elucidate the functional genes that regulate the spatiotemporal expression of capsaicinoids and explore the specific mechanisms underlying the synthesis of capsaicinoid-related genes.

The complex and sometimes contradictory influence of environmental factors on capsaicinoid content variation necessitates a thorough examination. Pepper varieties exhibit diverse responses to specific environmental conditions, a phenomenon also observed in other traits such as yield, fruit pigment, morphology, and vitamin C levels. The scarcity of research data in this particular field presents a challenge in deriving definitive conclusions from the available literature. It is probable that environmental factors exert their influence on capsaicinoids through various transcription factors. The predominant focus of the current research lies on the *MYB* family, underscoring the need for further investigations into other transcription factor families.

In order to enhance the comprehension of capsaicinoid metabolism, forthcoming studies may employ a multi-omics approach to identify additional transcription factors that regulate capsaicinoid metabolism. Simultaneous exploration of these factors can contribute to the development of a comprehensive and detailed capsaicinoid transcriptional regulatory network.

## Figures and Tables

**Figure 2 plants-13-02887-f002:**
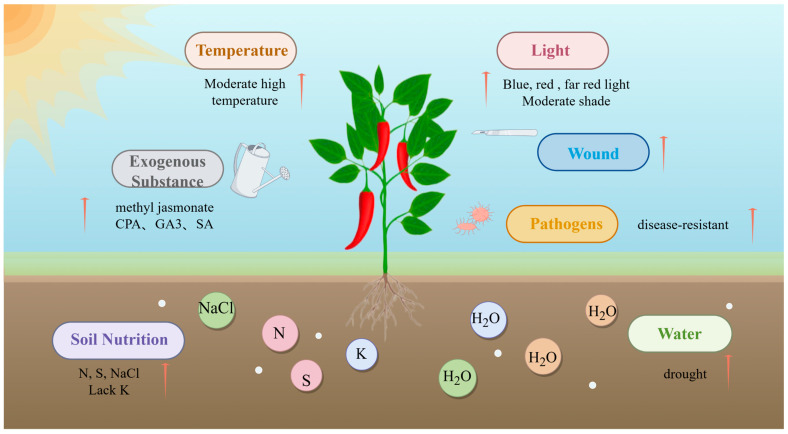
Effect of different environmental factors on capsaicinoids. Note: CPA: chlorophenoxyacetic acid; GA3: Gibberellic acid; SA: salicylic acid. Red arrows indicate different conditions promoting the synthesis of capsaicinoids, where the arrow in the pathogen module indicates that capsaicin improves disease resistance in chili.

## Data Availability

The original contributions presented in the study are included in the article, further inquiries can be directed to the corresponding author.

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
