# Peer review of "The Influence of Different Factors on the Metabolism of Capsaicinoids in Pepper (Capsicum annuum L.)"

_plants, 2024, doi:10.3390/plants13202887_

Round 1
Reviewer 1 Report
Comments and Suggestions for Authors
Overall, the review article discusses the influence of different factors on the metabolism of capsaicinoid in peppers. It may be best to include tables to help understand the text. Below are other broad suggestions:
A. Pay close attention to spacing within paragraphs, especially, at the end of sentences, after commas, etc.
B. Sentences should be written in active voice; several has been identified here.
C. Remove "s" from capsaicinoids especially when it is followed by the word "content"
D. Rephrase sentences with "process of" or "synthesis of" , use active voice
E. Clarify or rewrite lines 435-438
F. Check reference 69, 80, 91,97,98, 100-103, 107, 120,
G. References greater than 15 years old should be replaced.
H. Tables may be needed to help summarize previous studies.
Comments on the Quality of English Language
Quality of English is good.
Author Response
We have put the changes in word, please check it out.

Reviewer 2 Report
Comments and Suggestions for Authors
This review paper gives a very comprehensive coverage of our current understanding on the biosynthesis of capsinoids and their catabolism in pepper. Three groups of factors including genetic/developmental stage, environmental factors and chemicals are well covered, with possible mechanisms. Factors influencing at-harvest and post-harvest variations in capsinoids are also included. There are few of such review papers, hence this submission would have good general interests.
The shortcomings of the contribution are the lack of critical review and analysis, and the lack of summarising current knowledge in a concise manner. Figures and illustrations should be used more. Transcription factors and environmental factors and their target molecules/enzymes can be included in the same figure 1 or summarized in a different figure for easy reference.
Other suggestions for the authors to consider:
1. The paper is mostly about biosynthesis (anabolism) of capsinoids. It is understood that ‘metabolism’ is used in the title to include ‘catabolism (degradation, converting to other compounds). However, the continuous use of metabolism in the whole text can be confusing. The reviewer suggests using ‘biosynthesis’ and ‘degradation’ for the respective situations to differentiate anabolism and catabolism.
2. The whole paper takes capsinoids as a whole, seems to suggest that only the total quantity (not relative composition) is influenced by various factors. Authors need to double check if this is true and include those reports of differential responses in biosynthesis of different capsinoid molecules.
3. Multiple stress factors seem to enhance capsinoid biosynthesis, it is recommended to include plant health/productivity information if available to assess the possibility of adaptation of these stress treatment in farming.
4. Genome information on CBGs should also be discussed, especially information of the isoforms/homologues of the key CBGs. Do isoforms have different spatial and temporal expression patterns, and do they have different enzymatic activities?
5. In the review, there are multiple mention of ‘spiciness’, it should made clear if this ‘spiciness’ is the result of sensory testing or just another term to refer the total quantity of capsinoids.
6. Lane 128: should the pun1 gene be PUN1 gene?
Comments on the Quality of English LanguageMinor edit is recommended.
Author Response
We have put the changes in Word, please check it out.

Reviewer 3 Report
Comments and Suggestions for Authors
The present review: The influence of different factors on the metabolism of capsaicinoids in pepper (Capsicum annuum L.) summarizes many interesting findings regarding capsaicinoids, with particular emphasis on factors affecting their metabolism. On the other hand, I would recommend that the information on capsaicinoid biosynthesis be more detailed and careful. For example, the capsaicinoid biosynthetic pathway is discussed only in the form of a Fig. 1 legend, the key enzymes of biosynthesis are not discussed in the text. However, their introduction in the text is important in view of Section 4 discussing which factors influence/regulate the activity of the enzymes of biosynthesis. In addition, some names of enzymes and metabolites are inaccurate both in the legend and in the figure (I will clarify later). Another shortcoming is the lack of figures or summary tables. Fig. 1 is largely based on Naves et al.2019 Trends in Plant Science, Vol. 24. I was going to recommend add pictures of the fruits of the species listed on line 90 or the most important capsaicinoid structures, but even that is in the above mentioned publication. In your review, there should be more indication in which other publications all the capsaicinoid structures can be found, a diagram of a more detailed biosynthetic pathway (both phenylpropanoid part, branched-chain fatty acid pathway) or photos of other fruit species. On the other hand, your work is unique precisely because of the sum of external and internal factors influencing capsaicinoid biosynthesis, which you could document in an interesting figure.
Minor points:
writing the word chilli (14) x chili (26).
Abstract: the abstract is very brief, try to give specific examples e.g. on line 26 such as...
Fig. 1 p-coumaric instead of P-coumaric
p-coumaroyl-CoA instead of P-coumaroyl-CoA
p-coumaroyl-shikimate instead of P-coumaroyl-shikimate (please explaine more, is it the addition product of two phenolic acids?)
coenzyme A instead of coenayme A
Fig. 1 legend: l. 81 PAL: phenylalanine ammonia lyase instead of phenylalanine deaminase
l. 83: 3-hydroxylase instead of hydroxvlase
l. 83 caffeoyl-CoA 3-O-methyltransferase instead of coffee acyl-CoA 3-O-methyltransferase
l. 100 please explain the abbreviation of GWAS
l. 103 SNPs please explain
l. 135 analog alkaloid capsiate, the mention of analog alkaloid capsiate would deserve an explanation regarding biosynthesis, so is this another group of these substances, whose biosynthesis does not come from the phenylpropanoid pathway?
l. 171 and l. 185 Is it consistent with the citation rules to include the first names of authors of publications?
l. 178 The metabolism of capsaicinoids differs Please specify the title. Does it mean that the rate of formation of the final secondary metabolites is different, i.e. the activity of the individual biosynthetic enzymes difffers, or do the metabolic pathways differ directly, i.e. various enzymes and intermedates?
l. 192 Capsaicinoids may undergo degradation Degradation to which compounds?
l. 224 The content of capsaicinoids in peppers such as "Serrano", "Puya", "Anchoro", "Guajillo" and further in the text Which species or cultivars of which species?
l. 242 potentially due to the preservation of capsaicinoids enzyme activity. Please rephrase, capsaicins themselves do not have enzyme activity, they are not enzymes.
l. 255 GUS gene, please explain the abbreviation
l. 261 Isn't Hsp40 the preferred label for plant DnaJ? DnaJ is more for bacterial proteins
l. 355 What are black and red soils?
l. 387 what does Nigari contain?
l. 397 What is the causative agent of chili blight disease by its Latin name?
l. 399 Which NADH-dehydrogenase? Complex I of the respiration chain?
l. 406 (JA), a phytohormone, has been shown to be involved in capsaicinoid synthesis. Directly or as a signaling compound?
References: Citations in references are not uniform, e.g. abbreviations and full journal titles are used
Author Response

(The authors gave the same response as above.)

Round 2
Reviewer 3 Report
Comments and Suggestions for Authors
To my question:
Isn't Hsp40 the preferred label for plant DnaJ? DnaJ is more for bacterial proteins
We quite agree with you. However, a review of the references revealed that the authors used bioinformatics methods to identify putative DnaJ homology genes in capsicum.
But for proteins Fun et al. used both designation. Using only plant CaDnaJ protein is not appropriate.
Fig. 1 legend: Newly introduced abbreviations such as WRKY9, MYB31, bHL, H7,9, 26, 63, 86 are not explained
Minor points:
GUS gene is β-glu-curonidase gene, one word: β-glucuronidase gene
Abstract
Such as methyl jasmonate, chlorophenoxyacetic acid, Gibberellic acid, salicylic acid and other substances. It would be better to combine it with the previous sentence. Gibberelic why upper case letter G?
Author Response
We have submitted a coverletter for your review.
